# Selective HIF2A Inhibitors in the Management of Clear Cell Renal Cancer and Von Hippel–Lindau-Disease-Associated Tumors

**DOI:** 10.3390/medsci11030046

**Published:** 2023-06-30

**Authors:** Cristina Suárez, Maria Vieito, Augusto Valdivia, Macarena González, Joan Carles

**Affiliations:** Medical Oncology, Vall d’Hebron Institute of Oncology (VHIO), Vall d’Hebron Barcelona Hospital Campus, Hospital Universitari Vall d’Hebron, 08035 Barcelona, Spain

**Keywords:** VHL, HIF2A, clear cell renal cancer, Belzutifan, kidney cancer

## Abstract

Von Hippel–Lindau (VHL) loss is the hallmark event characterizing the clear cell renal cancer subtype (ccRCC). Carriers of germinal VHL mutations have an increased prevalence of kidney cysts and ccRCC as well as hemangioblastoma, pheochromocytoma and pancreatic neuroendocrine tumors. In both sporadic and inherited ccRCC, the primary mechanism of VHL-mediated carcinogenesis is the abnormal stabilization of hypoxia-inducible factors (HIF1A and HIF2A). While HIF1A acts as a tumor suppressor and is frequently lost through inactivating mutations/14q chromosome deletions, HIF2A acts as an oncogene promoting the expression of its target genes (VEGF, PDGF, CAIX Oct4, among others). Selective HIF2a inhibitors block the heterodimerization between HIF2A and ARNT, stopping HIF2A-induced transcription. Several HIF2A inhibitors have entered clinical trials, where they have shown a favorable toxicity profile, characterized by anemia, fatigue and edema and promising activity in heavily pretreated ccRCC patients. Belzutifan, a second-generation HIF2a inhibitor, was the first to receive FDA approval for the treatment of unresectable ccRCC in VHL syndrome. In this review, we recapitulate the rationale for HIF2a blockade in ccRCC, summarize the development of HIF2a inhibitors from preclinical models up to its introduction to the clinic with emphasis on Belzutifan, and discuss their role in VHL disease management.

## 1. Introduction

Renal cell cancer (RCC) is the 14th most common cancer worldwide, with an estimate of more than 431,000 cases and 180,000 cancer-related deaths occurring in 2020 [1]. The incidence of RCC peaks between 50 and 69 years of age, and males have a 70% higher risk of developing RCC [2]. The incidence of RCC is also increasing in the Western population, potentially due to lifestyle factors such as obesity, smoking and hypertension and the incidental detection of early stage tumors in abdominal imaging. 

The 2022 WHO classification has introduced several new types of RCC characterized by specific molecular alterations, but the clear cell subtype (ccRCC) is still by far the most prevalent subtype, accounting for 70% of cases [3]. Ninety-one percent of patients with ccRCC share the loss of a locus in the 3p chromosomal region that encompasses the VHL gene [4]. The complementary allele is inactivated through a combination of somatic or inherited mutations, hypermethylation and a loss of heterogeneity leading to biallelic loss of tumor suppressor genes in the vast majority of ccRCC cases [5,6]. Three tumor suppressor genes (BAP1, SETD2 and PBRM1) located in the same locus are also common targets of loss-of-function mutations [4]. 

In recent years, the small number of VHL wild-type tumors with clear cell cytoplasm have been mostly reassigned to other molecular subtypes such as MiT translocation renal cancer, TCEB1-mutated RCC and clear cell papillary renal cancer, and the existence of “true” ccRC tumors without VHL loss is controversial [5]. The central role of VHL in ccRC is supported by animal models that show the formation of renal cysts in animals with a loss of one VHL allele while those with biallelic loss develop RCC tumors [7], and by molecular analysis of multiple primary and metastatic samples that show that VHL loss is a clonal and early event in ccRC carcinogenesis [8].

## 2. HIF Pathway

The primary mechanism of VHL-mediated carcinogenesis is through the so-called hypoxia-inducible factors (HIF1A and HIF2A), which in normoxic conditions immediately undergo proteasome-mediated degradation after being ubiquitinated by VHL [9]. Only in hypoxic conditions do HIF1A and HIF2A stop being hydroxylated, rendering VHL unable to recognize them and exert its function and leading to abnormal stabilization, accumulation, and translocation of HIF1A/2A to the nucleus, where they interact with many genes modulating their expression in response to hypoxia. 

Some of these hypoxia-inducible genes are associated with increased proliferation (PDGFR and EGFR), angiogenesis (VEGF), glycolysis (CAIX and GLUT1), EMT (OCT4), and epigenetic deregulation (histone lysine demethylases) [10]. While the activation of those genes in response to hypoxia has a physiological role in helping repair the damage to the tissue and will eventually reverse when the normal levels of oxygen cause the inactivation of HIF1A/2A, tumors with VHL loss have a constitutive state of “pseudohypoxia”. However, VHL has more than 500 targets, including p53, RNA polymerase II subunits RBP1 and RBP7, and Spry2, that are independent of hypoxia, although the role of the non-HIF-mediated consequences of VHL loss in ccRC carcinogenesis is unclear [11].

In physiological conditions, HIF1A and HIF2A play complementary roles. HIF1A is expressed in response to hypoxia in almost any cell type, while HIF2A is only expressed by specific tissues, such as endothelial cells, cardiomyocytes, hepatocytes, adipocytes, neurons, and interstitial cells and glomeruli of the kidney [12], where HIF2A is key in the mediation of EPO production and iron metabolism.

## 3. HIF and RCC

There are, however, differences between HIF1A and HIF2A that point to opposite roles in RCC development. While HIF2A is almost invariably overexpressed in ccRCC samples, tumor microarray studies have confirmed that up to 40% of ccRCC is unable to produce HIF1A [13] due to deletions of the HIF1A gene situated in the 14q region in combination with inactivating mutations [14]. Downregulation of HIF1A in VHL mutant cells leads an increase in proliferation, suggesting that HIF1A acts as a tumor suppressor in ccRCC. HIF2A (but not HIF1A), or even that VHL is associated with worse prognosis in patients treated for ccRCC [15]. Some specific functions of HIF2A, not shared by HIF1A, include the activation of the Myc pathway, mTORC1, TGF-α and cyclin D1 [16]. 

Mutations in succinate dehydrogenase (SDH) and fumarate hydratase (FH), hypoxia-inducible factor prolyl hydroxylase 1/2/Egl-9 homolog (EGLN1/2), and the endothelial PAS domain-containing protein 1 (EPAS1) gene, which codifies the HIF2A protein, are all able prevent the hydroxylation of HIF and mimic the phenotype caused by the loss of VHL, resulting in tumors that will share many characteristics and therapeutic vulnerabilities [17]. 

## 4. Pre-Clinical Data with HIF Inhibitors 

While many of the treatments commonly used to treat ccRCC act over the downstream effects of VHL loss (VEGFR or PIK3K/mTOR overexpression), there is growing interest in developing specific inhibitors of HIF2A that could act upstream of the activation of hypoxia-modulated genes [17]. 

For many years, transcription factors such as HIF2A were considered to be undruggable, but the discovery of a pocket on the PAS protein–protein interaction domain led to the development of allosteric inhibitors that can bind specifically to the pocked and the heterodimerization with ARNT, which is needed before HIF2A can bind to its targets [18]. After screening a library of small molecules, two potential drug candidates were first generated, PT2399 and PT2385. PT2399 was validated in vivo, showing activity in both tumor cell lines and patient-derived xenografts, in both treatment-naïve and Sunitinib-resistant tumors, and was associated with a reduction in the levels of circulating EPO, a target of HIF2A in blood [19]. Likewise, PT2385 was also validated in multiple in vivo models, showing dramatic tumor responses in animal models and consistent dose-dependent inhibition of HIF2A targets, while no effect was observed in genes that are targets of HIF1A [20]. Unfortunately, in vivo studies showed a wide range of bioavailability with a significant number of patients not being able to achieve the target 500 ng/mL concentration of the drug in plasma [21]. 

A second generation of HIF2A inhibitors, showing improved pharmacokinetic profiles, selectivity, and potency, is represented by Belzutifan (also called MK-6482 and PT2977), DFF332 and NKT2152.

For example, Belzutifan showed AUC concentrations that were 6.9 times higher and 9.7 times the Cmax of PT2399. Unsurprisingly, not only did Belzutifan prove to be efficacious in xenograft models but the doses required were much lower than those of PT2399, leading to rapid regression of implanted tumors and the suppression of specific HIF2a targets, such as cyclin D1 [21], with much lower variability in the concentration of the drug in clinical studies. 

Likewise, in 2022, the preclinical studies for NKT2152 were communicated at AACR, showing dose-dependent disruption of the HIF2A/HIFB complex and inhibition of specific targets of HIF2A, such as VEGFA, cyclin D1, and GLUT1.

Interestingly, the preclinical activity of NKT2152 was not limited to VHL-deficient tumors, including, for example, hepatocarcinoma models. The authors suggest that HIF2A could play a role in modulating the microenvironment in hypoxic conditions frequently found in many tumor types [22]. 

Preclinical results for DFF332, another second-generation HIF2A, have not yet been published. 

## 5. VHL Disease

Scientific advances in the field of HIF2α would not have been possible without the previous discovery of von Hippel–Lindau (VHL) disease. VHL disease is an autosomal-dominant syndrome caused by pathogenic germline variants of the VHL gene. The incidence is 1 in 36,000 live births, and its penetrance is up to 90% [23].

VHL disease was first described by Eugel von Hippel and Arvid Lindau, who described the presence of highly vascular retinal and cerebellar tumors in young patients [24,25].

This disease is divided into two types: type 1 patients do not have pheochromocytomas, whereas type 2 patients almost invariably develop these tumors. Type 2 VHL disease is divided into 2A (with renal cancer), 2B (without renal cancer), and 2C (exclusively pheochromocytomas) [26].

The presence of highly vascularized benign and malignant tumors is a characteristic feature of this entity. Multiple tumors, including bilateral kidney tumors, frequently occur. The tumors occurring in this disease are summarized in Table 1.

Clinical suspicion of VHL disease warrants genetic counseling for VHL gene mutation screening. Genetic testing is recommended in all patients affected by retinal and craniospinal hemangioblastoma. Adults older than 50 with no other family history of VHL-related tumors should not undergo a genetic counseling consult [27]. 

When the diagnosis of VHL disease is established, secondary prevention programs are mandatory [28]. The recommended tests to be carried out are included in Table 2.

Surgery is the cornerstone treatment of VHL-disease-induced tumors. Radiation therapy and radiofrequency can also be viable options if resection is not feasible [29,30]. 

Systemic therapy is reserved for patients that cannot be managed with surgery, and tyrosine kinase inhibitors with antiangiogenic potential have been tested in this scenario; however, so far, none of them have achieved regulatory approval.

Two studies evaluated the efficacy of Sunitinib in tumors associated with VHL disease. The first was a single-arm, open-label, phase II clinical trial that enrolled patients with genetically confirmed VHL disease who had at least one measurable tumor associated with VHL disease [31]. The primary endpoint was safety, and the secondary endpoint was efficacy in terms of progression-free survival. Fifteen patients received four cycles of Sunitinib at a starting dose of 50 mg daily in cycles of 28 days, followed by 14 days of rest every six weeks. Major adverse events included fatigue (grade 3 in five patients), diarrhea, and mucositis. Dose reduction was required in 10 patients. The objective response rate was 33% (all renal cell carcinomas, no hemangioblastomas).

The PREDIR VHL network conducted an open-label, multicenter phase II that included patients with genetically confirmed advanced VHL disease [32]. Sunitinib was administered at the standard dose until disease progression. The primary endpoint was efficacy in terms of the objective response rate, and the secondary endpoint was safety and overall survival. Five patients were evaluated, and all patients achieved stable disease as the best response at six months, with two patients showing remarkable clinical improvement during the first few cycles of treatment. The overall survival was not reported due to the low incidence of events. Treatment-related adverse events were consistent with the expected tolerability profile of Sunitinib. However, unacceptable toxicities led to treatment discontinuation in three patients. The authors concluded that Sunitinib has limited benefit in patients with VHL disease. The other major TKI tested in VHL syndrome, Pazopanib, was studied in a nonrandomized, single-center, open-label phase II trial [33]. Patients with genetic or clinical features of VHL disease were treated with 800 mg of Pazopanib daily for 24 weeks. Primary endpoints included objective response and safety. Thirty-one patients received Pazopanib. The objective response was 42%. Treatment was discontinued in 13% of patients due to grade 3–4 transaminitis. Central nervous system hemorrhage (grade 5) occurred in one case.

## 6. Phase I Trials

Several phase I clinical trials have been conducted to explore the safety of HIF-2α inhibitors and set the preliminary efficacy data for phase II and III clinical trials being performed today. 

The first-in-human clinical trial of PT2385 included 51 patients with locally advanced or metastatic ccRCC (NCT02293980) [34]. PT2385 was administered orally at twice-daily doses of 100 to 1800 mg, according to a 3 + 3 dose-escalation design (26 patients), followed by an expansion phase at the recommended phase 2 dose (RP2D) with advanced ccRCC patients only (25 patients). Patients had a median age of 65 years (range, 29–80 years), 71% were male, 69% had an ECOG performance status score of 1, and 76% were considered at intermediate/poor risk by International Metastatic Renal Cell Carcinoma Database Consortium (IMDC) criteria. Patients received a median of four previous treatments; all patients received VEGF-targeted therapy, 61% received mTOR inhibitors, 39% received immune-checkpoint inhibitors, and 18% received cytokines. 

No DLTs were observed at any dose during the escalation cohort up to 1800 mg BID. Since the MTD was not reached, the RP2D was based on the observed safety, PK, and PD data and determined to be 800 mg twice a day. PT2385 was well tolerated, and the most common all-grade AEs were anemia (45%), peripheral edema (39%), and fatigue (37%). The most common grade ≥ 3 AEs were anemia (10%), lymphopenia (4%), and hypophosphatemia (8%). No patients discontinued treatment due to AEs, but five dose interruptions and two dose reductions were required. 

In terms of efficacy, the authors report an ORR of 14%, obtaining 1 CR (2%), 6 PR (12%), 26 SD (52%), and 17 PD (34%) cases. Twenty-one patients (42%) had SD or better for >4 months. The DCR was 66% with 13 patients remaining in the study for ≥1 year. 

Another phase I presented at ASCO 2019 was the combination of PT2385 with Nivolumab [35]. Patients were treated with PT2385 at 800 mg BID in combination with nivolumab at 3 mg/kg intravenously every 2 weeks to evaluate the safety, efficacy, and pharmacokinetics. A total of 50 patients with advanced ccRCC were enrolled; the median age was 62 years old, and 58% were ECOG 1. The median number of previous treatments was 1, and 42% of patients had received ≥2 prior lines of treatment. The most common all-grade AEs were anemia and fatigue (46% each), nausea (36%), and arthralgia (30%), and the most common grade 3 AEs were anemia and fatigue (4% each) and hypoxia (4%). Two grade 4 events were found, one due to increased ALT and another due to increased lipase. An ORR of 22% was observed with 1 CR and 10 PR. The median PFS for all patients after a median follow-up of 12.4 months was 7.3 m. A difference of 5.3 months in PFS was observed for those patients with sub-therapeutic exposures (<300 ng/mL) vs. therapeutic exposures of PT2385.

Belzutifan, a second-generation inhibitor with higher potency than PT2385, was tested in 95 patients in the phase I study LITESPARK-001 (MK-6482-001, NCT02974738) [36], including 43 patients with solid tumors (22 with ccRC 51%) treated at doses ranging from 20 mg to 240 mg QD and 52 patients with ccRC in a preplanned expansion cohort at an RP2D of 120 mg QD. The median age in the ccRC cohort was 63 years (range, 27–84 years), the majority of the patients (44 patients, 80%) were male, the ECOG status was 0–1 in 62%, and 76% were considered to be at intermediate/poor risk by International Metastatic Renal Cell Carcinoma Database Consortium (IMDC) criteria. Patients had received a median of three previous treatments; 50 patients (91%) received anti-VEGF agents; 44 patients (80%) received a checkpoint inhibitor; and 13 patients (24%) received a mammalian target of rapamycin (mTOR) inhibitor. Thirty-nine patients (71%) had received both anti-programmed death 1 and anti-VEGF agents. 

In the escalation phase, treatment-related adverse events (AEs) of any grade occurred in 42 of 43 patients (98%) and were considered treatment related in 72% of patients, but grade ≥ 3 treatment-related AEs only in only 19% of patients across dose levels. The most common reason for treatment discontinuation was progressive disease (60%). No dose-limiting toxicities were detected in the dose-escalation cohort at doses <160 mg QD. Treatment-related dose-limiting toxicities occurred in 14% of patients at 240 mg orally (grade 4 thrombocytopenia) and 17% of patients at 120 mg twice daily (grade 3 hypoxia). The maximum tolerated dose was not reached. No patients experienced grade 5 toxicity related to treatment. The RP2D was 120 mg once daily based on the safety, pharmacokinetics, and pharmacodynamics of the dose-escalation cohort. All patients in the ccRCC cohort treated at the RP2D presented at least one adverse event, mainly anemia (76%), fatigue (71%), dyspnea (49%), and/or nausea (36%). The most common grade 3 treatment-related adverse events (TRAEs) were anemia (27%) and hypoxia (16%). There were no G4-5 TRAEs in this study. No patients required dose reductions or discontinuation for anemia, but hypoxia led to dose interruption in six patient, dose reduction in two cases, and treatment discontinuation in two cases. 

With a median follow-up of 27.7 months when the data were reported, the objective response rate (ORR) was 25% (25% PR, 0% CR). Thirty patients (54%) experienced a best response of stable disease, providing a disease control rate (DCR) of up to 80%. Nineteen patients (35%) continued with Belzutifan for more than 12 months. The median duration of response was ≥6 months in 71% of patients. The ORR was 31% in patients with favorable risk and 24% in patients with intermediate/poor risk. The median progression-free survival for the overall cohort was 14.5 months (95% CI: 7.3 months—not reached). For patients with IMDC, a favorable median PFS was not reached, and for patients with IMDC, the intermediate/poor risk median PFS was 11 months. 

At the 3-year follow-up update, the authors reported a maintained ORR at 25%, with one confirming complete response (2%) and thirteen having partial responses (24%). No significant changes in adverse events were reported. The median duration of response was not reached (range from 3.1 to 37.9 months); 57% of responding patients remained in response at the data cut-off: July 2021. The DCR was 92% in those with a favorable IMDC risk and 76% for patients with an intermediate/poor risk. In patients who received prior VEGF/immunotherapy, an ORR of 21% and DCR of 94% were reported. The median PFS for the total cohort was 14.5 months (95% CI 7.3 to 22.1 months). No data on the overall survival have been reported. 

Other second-generation HIF-2α inhibitors such as DFF332 and NKT2152 have already entered clinical trials. A summary of ongoing phase 1 clinical trials can be found in Table 3. 

## 7. Role of Belzutifan in VHL Disease

The LITESPARK-004 trial (MK-6482-004) is an open-label, single-arm, phase II clinical trial designed to test Belzutifan in VHL disease. This clinical trial’s main purpose was to evaluate Belzutifan’s safety and efficacy in patients with VHL-associated renal cell carcinoma and other VHL-disease-associated tumors [37].

The patients included were more than 18 years old, had VHL disease diagnosed by germline VHL alteration, and had at least one measurable renal cell carcinoma tumor identified by a CT scan or MRI, defined according to RECIST version 1.1. Key exclusion criteria included tumors greater than 3 cm that required surgical intervention or metastatic disease. Belzutifan was administered at a dose of 120 mg daily. Treatment continued until disease progression or unacceptable toxicity. The primary endpoint was efficacy in terms of the objective response rate in patients with renal cell carcinoma. Key secondary endpoints included the safety, efficacy in terms of objective response rate in non-renal cell cancer neoplasms, and duration of response. A total of 61 patients were enrolled. Regarding the primary endpoint, the objective response rate was 49% (95% CI 36–62), all demonstrating partial responses. Thirty patients had stable disease as the best response. The median response time was 8.2 months (range 2.7–19.1). 

All patients had pancreatic lesions. Of these, the objective response was 77%, with six patients (10%) achieving a complete response. Patients with central nervous system hemangioblastomas had an objective response rate of 30%, which included three complete responses. 

The most common adverse events were anemia, fatigue, headaches, and dizziness. Treatment was interrupted in 34% of patients and discontinued in 15% due to adverse events. Importantly, all patients suffered from a reduction in hemoglobin levels during the first 13 weeks of treatment exposure. Treatment strategies for anemia control included blood transfusions, erythropoietin stimulating agents, or both. There were no treatment-related deaths reported in this trial. One patient suffered from grade 3 hypoxia that required dose interruption for one week, followed by dose reduction to 80 mg daily. This poorly understood phenomenon is likely related to an impaired pulmonary arterial vasoconstriction response induced by HIF2α inhibition [38]. 

Two patients had progressive disease as the best response; the mechanisms of possible resistance to Belzutifan have not yet been elucidated. LITESPARK-004 is the first trial showing clinical evidence of the activity of Belzutifan in treating hereditable cancer [39]. 

Based on these data, the FDA approved Belzutifan on 13 August 2021 as a first-in-class HIF inhibitor for adult patients with von Hippel–Lindau disease who require therapy for associated renal cell carcinoma, central nervous system hemangioblastomas, or pancreatic neuroendocrine tumors not requiring immediate surgery. 

At the ASCO 2022 annual congress, the authors presented an update with a median follow-up of more than two years. Disease progression of renal cell tumors occurred in four patients, but Belzutifan continued to show antitumor activity in VHL-disease-related neoplasms. Furthermore, the safety profile continued to be consistent with previous reports [39]. 

## 8. Other Phase 2 Clinical Trials

PT2385, a first-generation HIF2α inhibitor, showed in vivo activity against glioblastoma cells. In this regard, a two-stage, single-arm, open-label, phase II clinical trial was carried out and reported at the ASCO 2019 annual congress (NCT03216499) [40]. The patients included had their first disease recurrence after chemoradiation therapy. The primary outcome was the objective response, and secondary outcomes included the safety and overall survival. PT2385 was administered at a phase II dose of 800 mg BID. Twenty-four patients were enrolled in stage I. No objective response was observed, and the median progression-free survival was 1.8 months. Due to this, the trial was stopped. Important adverse events included hypoxia (3 patients), anemia (1 patient), and hyperglycemia (1 patient). 

Belzutifan combined with Cabozantinib is being explored in the LITESPARK-003 (NCT03634540), an ongoing, open-label, phase II trial [41]. The trial consists of two cohorts, treatment naïve (cohort 1) and previous treatment with immunotherapy (cohort 2). The primary endpoint was the objective response rate. Secondary endpoints include the duration of clinical benefit, progression-free survival, and overall survival. Patients included received Belzutifan 120 mg daily plus Cabozantinib 60 mg daily until disease progression or unacceptable toxicity. At the ESMO 2021 annual congress, the authors presented data from cohort 2. Patients with locally advanced or metastatic clear cell renal cell cancer that had received up to two previous treatment regimens were included. Fifty-two patients were enrolled in cohort 2. In the efficacy analysis, the objective response was 28.8% (all confirmed partial responses), and the median duration of response has not established. The median progression-free survival was 16.8 months (95%CI; 9.2—not reached), and the overall survival rate at 12 months was 81.3%. Most treatment-related adverse events were grade 1 or 2. Grade 3 adverse events included hypertension (23%), anemia (12%), fatigue (12%), and hypoxia (4%). There were no grade 4 or 5 adverse events at the presentation time. The authors conclude that the combination of Belzutifan with Cabozantinib shows a manageable safety profile and encouraging antitumor activity in previously treated advanced renal cell carcinoma. 

Furthermore, at the ESMO 2022 annual congress, the results from cohort 1 were presented. Thirty-five patients with untreated locally advanced or metastatic clear-cell renal cell cancer were included. At the time of data presentation, 35 of a planned 50 patients were enrolled. After a median follow up of 14 months, the efficacy outcomes were as follows: the objective response rate was 57% (6% of patients achieved complete response), and the DCR was 94%. The median time to response was 1.9 months (range: 1.7–9.2) while the median duration of response was 28.6 months (range: 1.7–28.6). The 12-month PFS and OS were 67% and 96%, respectively. Finally, the median progression-free survival was 30.3 months (95%CI, 9.4—not reached (NR)) and the median overall survival was NR. Concerning the safety, 97% of patients reported any-grade treatment-related adverse effect. However, only 37% of the recorded adverse events were grade 3, and no grade 4 or 5 adverse events were presented at data cut-off. As expected, the most common adverse events included anemia (all grade 71%), diarrhea (all grade 71%), and fatigue (all grades: 63%). Cabozantinib required dose reductions in 74% of patients, while Belzutifan required reductions in only 20%. Furthermore, Cabozantinib was discontinued in one patient, and no patient required discontinuation of Belzutifan due to toxicity. The authors concluded that dual targeting of HIF-2a and VEGF may be an effective treatment in advanced clear cell renal cell cancer, with phase III clinical trials on the move [41]. Data from cohort 2 have just been published with a median follow-up of 24.6 months. This cohort enrolled fifty-two patients with locally advanced or metastatic clear cell renal cell cancer that had received up to two previous treatment regimens. In the efficacy analysis, the objective response was 30.8%, 15 patients (29%) showed partial responses and 1 patient (2%) achieved a complete response, and the median duration of response was 18.6 months. The median progression-free survival and median overall survival were 13.8 months (95%CI; 9.2–19.4) and 24.1 months (95%CI; 20.0–37.4), respectively. Most treatment-related adverse events were grade 1 or 2. Grade 3 adverse events included hypertension (27%), anemia (15%), and fatigue (12%). One patient had a grade 5 AE (respiratory failure). The authors conclude that the combination of Belzutifan with Cabozantinib shows a manageable safety profile and encouraging antitumor activity in previously treated advanced renal cell carcinoma [21].

Other phase II and III clinical trials with Belzutifan and other novel HIF2α inhibitors are currently underway (as described in Table 3).

## 9. Conclusions

Our understanding of ccRCC has evolved dramatically over the last two decades since the discovery of the critical role of VHL and HIF2A in the carcinogenesis process in the vast majority of patients. This discovery has also helped elucidate the phenotype of patients with VLH syndrome and in the design of tailored HIF2A inhibitors. 

Several HIF2α inhibitors are under development, and their activity is being explored in ccRCC with promising results. To date, the most developed one is Belzutifan, which has already been approved by the FDA for patients with VHL disease and is currently being investigated in different settings of ccRCC disease; however, other second-generation HIF2A inhibitors are already in clinical development. 

Although it is outside the scope of this review, some preclinical results suggest that HIF2A inhibitors may have applications in other types of RCC in which the abnormal stabilization of HIF2A occurs due to mechanisms that are not caused by VHL loss, for example, in patients with SDH and FH mutations, and even more widely in tumors where the abnormal stabilization of HIF2A occurs as a consequence of prolonged hypoxia. 

## Figures and Tables

**Table 1 medsci-11-00046-t001:** Distribution of tumors that arise in VHL disease, according to system location.

Organ System	Manifestations
Nervous	Retinal and craniospinal hemangioblastomas, endolymphatic sac tumors (ELTS)
Genitourinary and adrenal	Renal cysts and renal cell cancer, pheochromocytoma, epididymal cystadenoma
Pancreatic	Cystadenomas, cysts, neuroendocrine tumors

**Table 2 medsci-11-00046-t002:** Summary of test recommended as follow-up for a patient with VHL disease.

Age (Years)	Test	Frequency	Disease
1	Ophthalmic examinations: direct and indirect ophthalmoscopy	Annual	Retinal hemangioblastoma
4	Blood pressure monitoring and 24 h urine studies for catecholamine metabolites. Alternatively, measure plasma-free metanephrines	Annual	Pheochromocytoma
12	Contrast-enhanced MRI brain and full spine	Annual or biennal	CNS hemangioblastoma
12	MRI examinations of the abdomen	Annual	Renal cell carcinoma and pancreatic neuroendocrine tumors
16	Audiogram	Biennial	Endolymphatic sac tumors

**Table 3 medsci-11-00046-t003:** A summary of ongoing phase 1 clinical trials.

Clinical Trial and Identifier	Treatment/s	Comparator/s	Primary Endpoint/s
Phase 1, NCT05030506 Advanced RCC	Belzutifan Pembrolizumab Lenvatinib	Two arms: Arm A: Belzutifan + Lenvatinib Arm B: Belzutifan + Lenvatinib + Pembrolizumab	DLTs, AEs, discontinuation due to AEs, AUC 0–24, Cmax, Tmax, Half-life, CL/F, Vz/F, AUC
Phase 1, NCT02974738 (MK-6482-001) Advanced solid tumors	Belzutifan	Two parts, four arms: Part 1A: Belzutifan all comers Part 1B: Belzutifan for advanced ccRCC Part 2: Belzutifan for other solid tumors Part 2A: Belzutifan for recurrent GBM who have been previously treated with CT/RT	MTD
Phase 1, NCT04846920 (MK-6482-018) ≥2nd Line ccRCC	Belzutifan	Four arms: Arm A: Belzutifan 160 mg BID Arm B: Belzutifan 160 mg TID Arm C: Belzutifan 200 mg BID Arm D: Belzutifan 120 mg QD	Percentage of AEs Discontinuation due to AEs Percentage of Participants Who Modify or Interrupt Study Treatment Due to an AE DLTs
Phase 1/1b, NCT04627064 Unresectable advanced or metastatic ccRCC (sarcomatoid included)	Abemaciclib Belzutifan	Two arms: Arm 1: Abemaciclib Arm 2: Abemaciclib + Belzutifan	Arm 1: ORR Arm 2: ORR and MTD
Phase 1/1b, NCT04895748 Advanced ccRCC and other malignancies with HIF2α stabilizing mutations (VHL, FH, SDHD, SDHAF2, SDHC, SDHB, SDHA EPAS1/HIF2A or ELOC/TCEB1)	DFF332 Everolimus Spartalizumab Taminadenant	Three arms: Arm 1: DFF332 Arm 2: DFF332 + Everolimus Arm 3: DFF332 + Spartalizumab + Taminadenant	Incidence and severity of AEs and SAEs Number of participants with dose reductions and interruptions Dose intensity for DFF332 for dose escalation and expansion Incidence of DLTs
Phase 1/2, NCT05119335 Advanced ccRCC who have exhausted available standard treatment (≤4 lines of treatment).	Oral NKT2152	Single arm	Phase 1: DLT and RP2D Phase 2: ORR
Phase 1/2, NCT05468697 Unresectable stage IV ccRCC > 2 lines of treatment (included IO and VEGF-TKI)	Palbociclib Belzutifan	Two parts: Phase 1: Belzutifan 1200 mg + Palbociclib at increasing doses (75, 100, 125 mg) Phase 2: Belzutifan 120 mg + Palbociclib at RP2D	Phase 1: DLTs, AEs, discontinuation due to AEs Phase 2: ORR
Phase 1/2, NCT04626479 (MK-3475-03A) 1st Line ccRCC	Belzutifan Pembrolizumab/Favezelimab Pembrolizumab/Quavonlimab Lenvatinib Vibostolimab/Pembrolizumab	Four arms: Pembrolizumab/Quavonlimab + Lenvatinib Pembrolizumab/Favezelimab + Lenvatinib Pembrolizumab + Belzutifan + Lenvatinib Pembrolizumab + Lenvatinib Vibostolimab/Pembrolizumab + Belzutifan	Safety Lead-in Phase: DLTs, AEs, discontinuation due to AEs. Efficacy Phase: DLTs, AEs, discontinuation due to AEs, and ORR.
Phase 1/2, NCT04626518 (MK-3475-03B) ≥2nd Line ccRCC	Belzutifan Pembrolizumab/Favezelimab Pembrolizumab/Quavonlimab Lenvatinib MK-4830	Six arms: Pembrolizumab/Quavonlimab Pembrolizumab/Favezelimab Pembrolizumab + MK-4830 Pembrolizumab + Belzutifan Belzutifan + Lenvatinib Pembrolizumab + Lenvatinib	Safety Lead-in Phase: DLTs, AEs, discontinuation due to AEs. Efficacy Phase: DLTs, AEs, discontinuation due to AEs, and ORR.
Phase 2, NCT03634540 (MK-6482-003) Advanced or metastatic ccRCC	Belzutifan Cabozantinib	Two cohorts: Cohort 1: Belzutifan + Cabozantinib in treatment naïve Cohort 2: Belzutifan + Cabozantinib: Prior IO	ORR
Phase 2, NCT04489771 (MK-6482-013) Advanced or metastatic ccRCC	Belzutifan	Dose A (standard dose 120 mg) Dose B (higher dose)	ORR
Phase 3, NCT04195750 (MK-6482-005) Unresectable, locally advanced or stage IV ccRCC (with PD after IO and VEGF-TKI in sequence or in combination and no more than three lines of treatment)	Belzutifan Everolimus	Two arms: Experimental: Belzutifan 120 mg QD Control: Everolimus 10 m QD	PFS by BICR OS (from randomization)
Phase 3, NCT05239728 (MK-6482-022) ccRCC (with or without sarcomatoid features) Intermediate-high risk, high risk, or M1 no evidence of disease (NED)	Belzutifan Pembrolizumab Placebo	Two arms: Experimental: Pembrolizumab + Belzutifan Control: Pembrolizumab + placebo	DFS
Phase 3, NCT04736706 (MK-6482-012) 1st Line ccRCC	Belzutifan Pembrolizumab Pembrolizumab/Quavonlimab Lenvatinib	Two arms: Experimental: Belzutifan + Pembrolizumab + Lenvatinib Experimental: Pembrolizumab/Quavonlimab + Lenvatinib Control: Pembrolizumab + Lenvatinib	PFS by BICR OS (from randomization)
Phase 3, NCT04586231 (MK-6482-011) 2nd/3rd Line ccRCC after PD-1/L1	Belzutifan Lenvatinib Cabozantinib	Two arms: Experimental arm: Belzutifan 120 mg + Lenvatinib 20 mg QD Control: Cabozantinib 60 mg QD	PFS by BICR OS (from randomization)

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
