# Peer review of "Selective HIF2A Inhibitors in the Management of Clear Cell Renal Cancer and Von Hippel–Lindau-Disease-Associated Tumors"

_medsci, 2023, doi:10.3390/medsci11030046_

Round 1
Reviewer 1 Report
I have little comments on the comprehensive review by Suarez et al:
- I find the chapter 5 on VHL disease (syndrome) a bit unnecessary and making the review unnecessary long as most of the rest deals with sporadic ccRCC
- Since belzutifan (and its predecessor) is so far the only HIF-2a inhibitor with clinical trial results, the title could be …HIF2a inhibitor
- line 19: … ccRCC in VHL syndrome.
- In section 8 “the cohort 2” results are described in two different paragraphs. Should be combined in one.
Author Response
- I find the chapter 5 on VHL disease (syndrome) a bit unnecessary and making the review unnecessary long as most of the rest deals with sporadic ccRCC:
Thank you for the feedback, since the first approved indication of HIF2 inhibitors is the treatment of patients with VHL syndrome we think it is valuable to familiarize the intended audience with this disease.
We will review this section to highligth practical issues and consider shortening it.
- Since belzutifan (and its predecessor) is so far the only HIF-2a inhibitor with clinical trial results, the title could be …HIF2a inhibitor
The review includes an extensive discussion of other HIF2a inhibitors, so the title is actually appropiate.
- line 19: … ccRCC in VHL syndrome. Corrected.
- In section 8 “the cohort 2” results are described in two different paragraphs. Should be combined in one. Corrected.
Reviewer 2 Report
The paper is well organised. The VHL deficiency is known for the clear cell kidney cancer as an independent prognostic markers. Its heterogeneity becomes the origin of the low effectiveness of the anti-cancer therapy.
New insight on the anti-angiogenic therapy will provide the better reponse to the treatment and higher common survival rates.
English language is good for the paper
Author Response
Thank you very much for your review